# Bacterial and Cellular Response to Yellow-Shaded Surface Modifications for Dental Implant Abutments

**DOI:** 10.3390/biom12111718

**Published:** 2022-11-20

**Authors:** Tullio Genova, Giorgia Chinigò, Luca Munaron, Paola Rivolo, Anna Luganini, Giorgio Gribaudo, Davide Cavagnetto, Pietro Mandracci, Federico Mussano

**Affiliations:** 1Department of Life Sciences and Systems Biology, University of Turin, 10123 Turin, Italy; 2Department of Applied Science and Technology, Materials and Microsoystems Laboratory (ChiLab), Politecnico di Torino, 10129 Torino, Italy; 3Department of Surgical Sciences, CIR Dental School, University of Turin, 10126 Turin, Italy; 4Politecnico di Torino, 10129 Torino, Italy

**Keywords:** dental implants, surface modification, mucosal seal

## Abstract

Dental implants have dramatically changed the rehabilitation procedures in dental prostheses but are hindered by the possible onset of peri-implantitis. This paper aims to assess whether an anodization process applied to clinically used surfaces could enhance the adhesion of fibroblasts and reduce bacterial adhesion using as a reference the untreated machined surface. To this purpose, four different surfaces were prepared: (i) machined (MAC), (ii) machined and anodized (Y-MAC), (iii) anodized after sand-blasting and acid etching treatment (Y-SL), and (iv) anodized after double acid etching (Y-DM). All specimens were characterized by scanning electron microscopy (SEM) and energy-dispersive X-ray spectroscopy (EDX). Moreover, the mean contact angle in both water and diiodomethane as well as surface free energy calculation was assessed. To evaluate changes in terms of biological responses, we investigated the adhesion of *Streptococcus sanguinis* (*S. sanguinis*) and *Enterococcus faecalis* (*E. faecalis*), fetal bovine serum (FBS) adsorption, and the early response of fibroblasts in terms of cell adhesion and viability. We found that the anodization reduced bacterial adhesion, while roughened surfaces outperformed the machined ones for protein adsorption, fibroblast adhesion, and viability independently of the treatment. It can be concluded that surface modification techniques such as anodization are valuable options to enhance the performance of dental implants.

## 1. Introduction

In recent decades, dental implants have revolutionized prosthetic dentistry owing to a previously unknown intimate interaction between bone and titanium fixtures [1]. The presence of a trans-mucosal component implies an interface between titanium and gingival cells such as fibroblasts and epithelial cells that may require surface features different from those optimized for intraosseous usage [2]. Moreover, clinical responses such as severe gingival recession and so-called peri-implantitis have recently highlighted the importance of soft tissue sealing around implants as a possible barrier to bacterial penetration along the fixture [3,4]. Unfortunately, attaining this sealing is not an easy task, as the periodontium is characterized by a very distinctive connective tissue, the periodontal ligament, and the epithelium normally adhering to the surface of a natural tooth [5] acts differently around implants [6,7]. There are indeed several limitations regarding the height and quality of the soft tissue surrounding an implant, although the extra-osseous portion of an implant can be designed to reduce plaque accumulation [8]. To ameliorate soft tissue attachment, several approaches have been proposed since the pivotal study by Abrahamsson et al. [9].

In this context, yttria-stabilized zirconia has been introduced as a viable option claiming a more favorable mucosal seal, i.e., elongated peri-implant epithelium, compared to conventional titanium implants [10]. A remarkable number of studies have instead focused on improving—through surface modification techniques [11]—the biological response of titanium, which benefits from undoubtedly superior mechanical features rather than massive ceramic materials. Grafting polylysine homopolymers has emerged as a promising strategy even for mucous tissues based on preliminary in vitro reports [12]. With a similar positive outcome, the UV light increased significantly the adherence to the surfaces of both adult mucosal and embryonic fibroblasts compared to the untreated control, possibly due to the enrichment with TiOH molecules [13]. The effect of UV light has been compared to argon plasma with regard to the interaction of fibroblasts with different abutment surfaces, with the latter outperforming the former [14]. The long-term efficacy of argon plasma treatment remains, however, unclear [15].

Differently from the above-mentioned approaches that are either still at an early stage of development or supported by poor clinical evidence, the anodization process, widely used to engineer the intra-bony surface oxide layer, has been suggested to guide the selective adhesion of the fibroblasts [16,17] and possibly bacterial adhesion trough the rutile/anatase phase tuning [18]. Consistently, novel implant platforms were introduced, in 2019 [19], based on an anodization gradient form the apical to the extraosseous part of the device. Since controversial data about the role of roughened surfaces are reported in the literature [20,21,22,23] as for the soft tissue response, in this work, we aimed to assess the effects of the anodization process applied to both machined and roughened surfaces in terms of both fibroblast cellular responses and bacterial adhesion. To this purpose, the early response of fibroblasts in terms of cell adhesion and viability was investigated. Furthermore, the protein adsorption and the adhesion of two bacterial strains, i.e., *Streptococcus sanguinis* (*S. sanguinis*) and *Enterococcus faecalis* (*E. faecalis*), were evaluated.

## 2. Materials and Methods

### 2.1. Sample Preparation

Commercially pure grade IV titanium samples were shaped into 12 × 4 mm cylinders (2r × h). Four types of specimens were prepared: (i) machined (MAC), (ii) machined and anodized (Y-MAC), (iii) anodized after sand-blasting and acid etching treatment (Y-SL), and (iv) anodized after double acid etching (Y-DM). The samples were cleaned with acetone and rinsed in a 70% isopropanol aqueous solution; subsequently, they were decontaminated in an ultrasound bath for 5 min in isopropanol and rinsed in Milli-Q water (Millipore, Billerica, MA, USA). To generate the anodized layer, Y-MAC, Y-SL, and Y-DM were immersed in a galvanic cell containing a solution of phosphoric acid and trisodium phosphate at a voltage of 50 ± 10 V for 10 min (Titanmed, Galbiate, Italy). The anodization process is known to produce a Ti oxide coating that is thicker than the one spontaneously formed on Ti samples exposed to atmospheric oxygen. This thicker Ti oxide coating appears yellow at the parameters used here (hence the use of Y in the acronyms). At the end of the process, the samples were washed in Milli-Q water for 20 min.

### 2.2. Scanning Electron Microscopy (SEM) and Energy-Dispersive X-ray Spectroscopy (EDX)

The details about surface morphology were captured using scanning electron microscopy (Phenom XL G2 Desktop SEM, Thermo Fisher Scientific Inc., Waltham, MA, USA) with an accelerating voltage of 20 kV and a magnification of 730× and 770× for each sample. In addition to the images, an energy-dispersive X-ray analysis was performed with the same instrument set at a magnification of 770× and 20 kV of voltage.

### 2.3. Contact Angle and Surface Energy Evaluation

Wetting properties were investigated by optical contact angle (OCA) measurements with the sessile drop technique, using an OCAH 200 (DataPhysic Instruments GmbH, Filderstadt, Germany). Water (dH_2_O) and diiodomethane (CH_2_I_2_) were used as probes. Each liquid drop (1 μL in volume) was dispensed, and the image of the drop on the sample was acquired with the integrated high-resolution camera. The drop profiles were extracted and fitted with dedicated software (SCA20) through the Young–Laplace method, and contact angles, at the liquid–solid interface, between the fitted function and baseline were calculated. For each sample and each liquid probe, the contact angle measure was repeated five times on different areas. Polar and dispersive components of the surface energy were finally estimated by applying the Owens–Wendt method [24], starting from the average contact angle estimated for each of the two different liquid probes.

### 2.4. Bacterial Biofilm Evaluation

The sterilized titanium disks were colonized by *S. sanguinis* and *E. faecalis*. Bacteria were grown overnight in 10 mL of Mueller Hinton (MH) broth (Sigma Aldrich, St. Louis, MO, USA) at 37 °C in agitation. The day after, bacteria were subcultured until an optical density (OD_600_) of 0.6 was reached, corresponding to 1 × 10^8^ colony-forming units (CFU)/mL, approximately. Each disk was incubated with 1 mL of MH broth (as a negative control) or 1 mL of bacterial suspension in a 24-well plate by using a shaking rotator (80 rpm) at 37 °C for 24 h. To remove non-adherent bacteria, each disk was rinsed in sterile saline solution and vortexed for 10 s, six times. Discs were then transferred into a sterile plastic container with 1 mL saline solution and sonicated at 80 kHz with a power output of 250 W. Afterwards, 10-fold dilutions of each supernatant were plated in the MH plate for colony counting [25].

### 2.5. Protein Adsorption

To quantify the amount of protein adsorbed onto the titanium disks, specimens were incubated in the presence of fetal bovine serum (FBS) (2% in phosphate buffered saline (PBS 1×) at 37 °C for 30 min, then washed twice with PBS. The total adsorbed protein amount was first eluted from the samples with Tris Triton buffer (10 mM Tris (pH 7.4), 100 mM NaCl, 1 mM EDTA, 1 mM EG-TA, 1% Triton X-100, 10% glycerol, and 0.1% SDS) for 10 min, and then quantified by means of a Pierce™ BCA Protein Assay Kit (Life Technologies, Carlsbad, CA, USA) according to the manufacturer’s instructions.

### 2.6. Cell Culture

The fibroblast cell line (NHDF, ECACC, Salisbury, UK) was used to assess the biocompatibility of the surface treatments [26]. As previously reported [14,27,28], cells were maintained in an incubator at 37 °C, in growth medium DMEM supplemented with 10% FBS (Life Technologies, Milan, Italy), 100 U/mL penicillin, and 100 μg/mL streptomycin, under a humidified atmosphere of 5% CO_2_ in the air.

### 2.7. Cell Adhesion

Cells were maintained and manipulated as reported in detail elsewhere [28]. Briefly, cells were detached using trypsin for 3 min, carefully counted, and seeded at 3 × 10^3^ cells/disk in 100 μL of growth medium on the different samples. Samples were kept in an incubator at 37 °C for 10 min, and then fixed by using 4% paraformaldehyde in PBS. Cells’ nuclei were stained with DAPI and counted on previously captured pictures, following established protocols [27,29].

### 2.8. Cell Proliferation

In order to evaluate proliferation, cells were plated at a density of 2500 cells/sample in 24-well culture dishes, and the proliferation was assessed by measuring luminescence through the commercial kit “Cell Titer GLO” (Promega, Madison, WI, USA) according to the manufacturer’s protocol 24 h after plating [30,31].

### 2.9. Statistics

Data are expressed as mean ± SEM. Statistical analyses were performed using GraphPad Prism software (Graph Pad Software Inc., San Diego, CA, USA), and differences with a *p*-value <0.05 were considered statistically significant (*: *p* ≤ 0.05). Statistical significance between different conditions was determined by analysis of variance (ordinary one-way ANOVA test) to compare more than two conditions to each other (in bacterial biofilm evaluation). The Mann–Whitney test was used to evaluate significance between two different conditions within one experiment (protein adsorption, cell adhesion, and cell proliferation assays).

## 3. Results

### 3.1. SEM

As shown in Figure 1, the scanning electron microscope (SEM) analysis performed on the titanium samples revealed a flat surface topography for the machined (MAC) titanium (Ti) specimens characterized by the typical finishing of the milling procedure. The same topography was also observed in anodized samples (Y-MAC), indicating that the thin coating generated through anodization cannot alter these characteristics (Figure 1). As regards Y-SL and Y-DM samples, rough surfaces typical of the subtractive process were observed (Figure 1).

### 3.2. EDX

To evaluate the chemical composition of titanium samples, EDX analysis was performed. As can be appreciated from Table 1, in all anodized samples (Y-MAC, Y-SL, and Y-DM), the amount of oxygen is higher, suggesting a greater amount of titanium compared to the untreated machined surface (MAC).

### 3.3. Wetting Properties

The wetting properties of MAC and anodized samples (Y-MAC, Y-SL, and Y-DM) were evaluated by measuring the optical contact angle (OCA) of water (H_2_O) and diiodomethane (CH_2_I_2_). MAC samples showed hydrophilic behavior as previously shown [28], with an average contact angle (CA) value of ~35° for water and ~40° for diiodomethane (CH_2_I_2_) (Figure 2A). The anodization process seems to decrease the hydrophilic properties of the specimens, as the mean contact angle measured for water was 93°, 106°, and 110° for Y-MAC, Y-SL, and Y-DM, respectively (Figure 2A). In addition, polar and dispersive components of the surface free energy (SFE) were calculated starting from the CA values of water and diiodomethane according to the Owens–Wendt theory. In accord with the previous observation, we found that anodization markedly reduced the SFE completely abolishing its polar component (Figure 2B).

### 3.4. Evaluation of Bacterial Biofilm

Biomaterial structural features may affect the bacterial biofilm forming around the dental implant. For this reason, *S. sanguinis* and *E. faecalis* were incubated in the presence of the different samples, and the adherent colonies were quantified as reported in Figure 3. A statistically significant reduction in the number of bacteria was found for Y-MAC, Y-DM, and Y-SL compared to MAC.

### 3.5. Evaluation of Biological Responses

The adsorption of protein on biomaterials surfaces is known to be involved in the cellular response to the biomaterial. Therefore, we evaluated the ability of the anodization process and the acid etching treatment to affect the adsorption of protein on the surface of titanium samples by performing protein adsorption assays. As shown in Figure 4, anodization increases the amount of proteins adsorbed on the different surface types. More specifically, we observed that combining anodization with increased surface roughness significantly improves protein adsorption on biomaterials. Indeed, Y-SL and Y-DM displayed a significantly higher level of proteins adsorbed compared to MAC.

Next, to assess how anodization and acid etching influence the early stages of interaction with the cells, cell adhesion experiments were carried out. As shown in Figure 5A, no significant differences between MAC and Y-MAC were observed. By contrast, consistent with the previous results obtained on protein adsorption (Figure 4), Y-SL and Y-DM significantly increased the number of adherent cells at 10 min, confirming a positive role of acid etching and the consequent increase in surface roughness in improving the biological performance of such titanium-based biomaterials.

Finally, the effect of anodization on cell proliferation was investigated. Also in this case, the anodization of the MAC surfaces did not show any effect on fibroblast proliferation 24 h after seeding, while it is possible to appreciate a significant increase in cell proliferation in Y-DM and Y-SL samples (Figure 5B).

## 4. Discussion

The so-called mucosal seal is believed to have a major role in preventing the onset of periodontitis, which is the major cause of implant failure [32]. Therefore, growing interest has developed toward the soft tissue responses elicited by the surface modifications of dental implants [11,33,34,35]. The peri-implant mucosa responsible for the soft tissue sealing exerts this function mainly owing to its connective lamina, whose major cell population is fibroblasts [36]. Therefore, these cells become central for achieving the correct integration of dental implants to the adjacent gingiva [37], a process in which the early cell response is driven by surface properties [38,39]. Surface modifications specifically targeting the attachment and viability of fibroblasts are recent [34,40,41].

In this study, two roughened surfaces, representative of the most common implant systems and obtained respectively by sand blasting–acid etching (SL) and double acid etching (DM) with subsequent anodization were compared with the anodized machined surface (Y-MAC). The anodization process was performed according to the current standards adopted in the implant industry to prepare yellow titanium specimens. Indeed, an oxide coating can refract and absorb light, generating inference colors based on its thickness [28]. Non-anodized machined surfaces (MAC) were analyzed as controls. The SEM images revealed the typical expected topography for the surface of the investigated samples (Figure 1), and the EDX confirmed a greater amount of oxygen on the anodized surfaces compared to the non-anodized MAC (Figure 2).

Concerning the biological characterization of the samples, we found that roughened surfaces (Y-DM and Y-SL) significantly increased protein adsorption, as well as fibroblasts’ adhesion and proliferation, compared to the MAC surfaces. These results are in agreement with several data previously published establishing a clear correlation between protein adsorption and early cell behavior [42]. To note, all the effects mediated by roughened surfaces resulted independent of the anodization treatment as MAC and Y-MAC displayed similar behavior. Conversely and quite interestingly, the anodization process revealed a significant impact on the adhesion of bacteria like *S. sanguinis* and *E. faecalis*. All anodized samples (Y-MAC, Y-SL, and Y-DM) showed a significant reduction in bacterial adhesion compared to MAC. This finding could be related to the surface free energy and its composition (polar vs. dispersive component) rather than to the surface topography. Indeed, all anodized samples are characterized by an SFE mainly associated with a high dispersive component unlike MAC surfaces, which show a higher SFE resulting from both polar and dispersive components.

The high bacterial adhesion observed in anodized specimens is in accordance with the linear relationship between high SFE and the number of adherent bacteria reported in the literature [43]. Furthermore, it appears that marked hydrophobicity reduces bacterial adhesion [44,45], while up to now no dependence between bacterial adhesion and topographical characteristics has been found [46,47].

The matter becomes even more complex if we consider that the SFE of a given bacterial strain determines its surface adhesion according to Ahn et al. [48]. The choice of the bacteria tested was not accidental. In fact, subgingival plaque containing *S. sanguinis* has been frequently retrieved from peri-implantitis, in which, despite a large number of dysbiotic species, *E. faecalis* was considered among the primary enablers [49]. Current knowledge has progressed incredibly since the previous assumption of a substantial identity between the flora of periodontal diseases and peri-implantitis. In this context, it has been demonstrated not only that the periodontitis microbiome significantly differs from that of peri-implantitis [50], but also that Ti particles can deeply influence the peri-implant microbiota [51]. In this regard, Daubert et al. underscored the role of tribocorrosion in reducing the maintenance of a healthy balance between the medical device and recipient [51]. However, further investigations should be focused on the study of the human microbiota in the presence of other abutment modifications to assess the clinical efficacy of this technology. In this context, the combination of the anodization process herein proposed and a barrier coating to prevent Ti ion diffusion from the bulk material of the fixture [52] could be taken into consideration to reduce bacterial adhesion and help select less aggressive microbiota.

## 5. Conclusions

The anodization of Ti specimens reduces bacterial adhesion, whereas roughened surfaces obtained by acid-etching treatments outperformed the machined ones in terms of both protein adsorption and cell response independently of the anodization process. Thus, it can be concluded that surface modification techniques as proposed are valuable options to enhance the performance of dental implants.

## Figures and Tables

**Figure 1 biomolecules-12-01718-f001:**
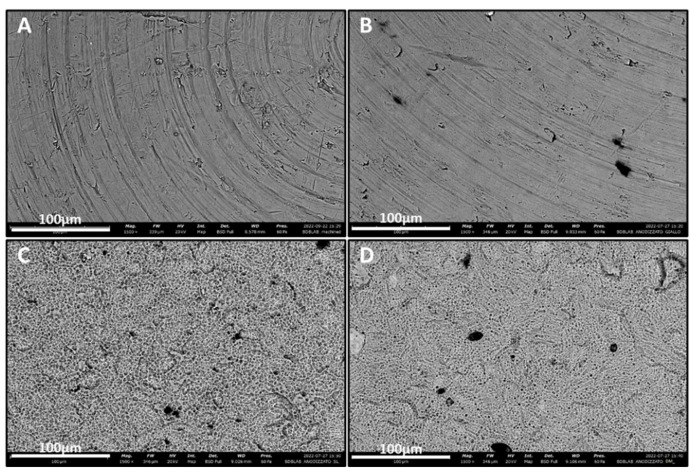
SEM images showing the surface topography of the titanium samples at high magnification (1500×). (**A**) MAC; (**B**) Y-MAC; (**C**) Y-SL; (**D**) Y-DM.

**Figure 2 biomolecules-12-01718-f002:**
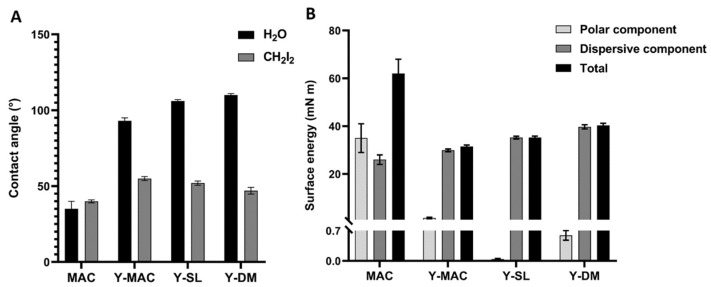
(**A**) Contact angle. Water (H_2_O) and diiodomethane (CH_2_I_2_) contact angles measured on MAC, Y-MAC, Y-SL, and Y-DM surfaces; (**B**) Surface energy. Contribution of polar and dispersive components and total contribution to surface energy, calculated by the Owens–Wendt method. Data are expressed as mean ± SEM.

**Figure 3 biomolecules-12-01718-f003:**
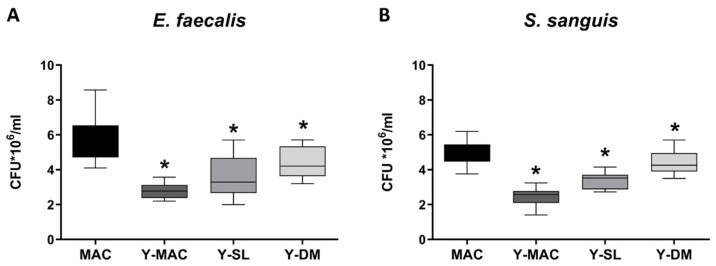
Biofilm quantification of *E. faecalis* (**A**) and *S. sanguinis* (**B**) strains on titanium samples. Data are displayed as mean ± SEM. Statistical significance versus MAC *: *p* < 0.05 (ordinary one-way ANOVA).

**Figure 4 biomolecules-12-01718-f004:**
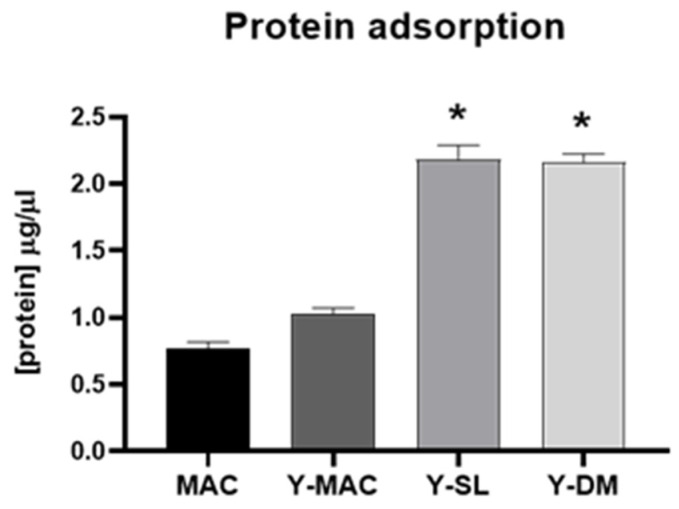
Protein adsorption. Quantification of FBS adsorbed on MAC, Y-MAC, Y-SL, and Y-DM. Data are displayed as mean ± SEM and refer to four independent experiments. Statistical significance versus MAC *: *p* < 0.05 (Mann–Whitney test).

**Figure 5 biomolecules-12-01718-f005:**
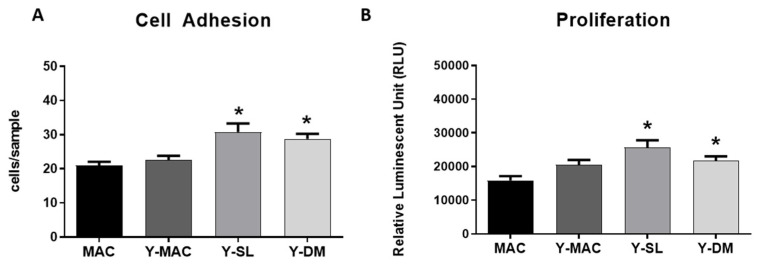
Quantification of cell adhesion (**A**), and cell proliferation (**B**) assays performed on MAC, Y-MAC, Y-SL, and Y-DM. Data are displayed as mean ± SEM and refer to four independent experiments. Statistical significance versus MAC *: *p* < 0.05 (Mann–Whitney test).

**Table 1 biomolecules-12-01718-t001:** Chemical composition of titanium samples obtained through EDX analysis.

Element			Atomic Conc.	Weight Conc.
Atomic Number	Symbol	Name		
			**MAC**	
8	O	Oxygen	7.670	2.700
22	Ti	Titanium	92.330	97.300
			**Y-MAC**	
8	O	Oxygen	57.012	30.700
22	Ti	Titanium	42.988	69.300
			**Y-SL**	
8	O	Oxygen	50.478	25.400
22	Ti	Titanium	49.522	74.600
			**Y-DM**	
8	O	Oxygen	49.412	24.600
22	Ti	Titanium	50.588	75.400

## Data Availability

All data are archived in Via Accademia Albertina 13, Torino 10123 IT ad DBIOS.

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
