# Peer review of "Bacterial and Cellular Response to Yellow-Shaded Surface Modifications for Dental Implant Abutments"

_biomolecules, 2022, doi:10.3390/biom12111718_

Round 1

Reviewer 1 Report

The manuscript presents a study regarding anodized titanium specimens and acid-etching treatments that can be used to reduce the bacterial adhesion and enhance the adhesion of fibroblasts, with application in dental implant abutments.

Overall, the manuscript is well written and well organized.

However, some revisions should be made:

-          The mention in the title of “gold-shaded surface modifications” might be confusing for readers, since “gold” is usually used to denote the noble metal (Au). Therefore, is should be changed with something more appropriate. In the manuscript, there was made only one reference to “yellow titanium specimens”, when this must be better described under “Methods”.

-          The introduction must be improved by providing more information about the state-of-the-art relevant to the amelioration of the attachment of soft tissues to implant abutments and the problems that are still unresolved. The motivation of this study, in this form, is not very convincing in term of originality (what is new as compared to other studies?). Moreover, the references mentioned in this section are rather old. Aren’t more recent studies published in this field?

-          Under section 2 (“Materials and Methods”), it must be shortly described the statistics method used.

-          The scale bar on SEM images is not visible (it is too small).

-          Lines 74-76: more parameters should be provided about the anodization process, such as for how long the samples were subjected to this process and the current/voltage that was used. The mention “at a given energy intensity” is too vague.

-          In table 1, the first column should be renamed “Atomic number” (not just “Number”) to avoid any confusion. From the EDX analysis, besides oxygen and titanium, no other element was present?

-          The images from Figs. 2 and 3 should be provided at a better resolution.

Author Response

-          The mention in the title of “gold-shaded surface modifications” might be confusing for readers, since “gold” is usually used to denote the noble metal (Au). Therefore, is should be changed with something more appropriate. In the manuscript, there was made only one reference to “yellow titanium specimens”, when this must be better described under “Methods”.

Thank you for your comments. We performed some amendments in the title and improved the description of the anodization process under the section "methods"

-          The introduction must be improved by providing more information about the state-of-the-art relevant to the amelioration of the attachment of soft tissues to implant abutments and the problems that are still unresolved. The motivation of this study, in this form, is not very convincing in term of originality (what is new as compared to other studies?). Moreover, the references mentioned in this section are rather old. Aren’t more recent studies published in this field?

The introduction was amended as suggested by providing more information about the
state-of-the-art regarding the mucosal seal and the motivation of the study. Finally, more
recent references were included.

-          Under section 2 (“Materials and Methods”), it must be shortly described the statistics method used.

"Statistics" paragraph was included at the end of methods section 

-          The scale bar on SEM images is not visible (it is too small).

The scale bar was enlarged according to your suggestions

-          Lines 74-76: more parameters should be provided about the anodization process, such as for how long the samples were subjected to this process and the current/voltage that was used. The mention “at a given energy intensity” is too vague.

The authors agree and amended the text as required.

-          In table 1, the first column should be renamed “Atomic number” (not just “Number”) to avoid any confusion.

The authors apologize for the omission.

From the EDX analysis, besides oxygen and titanium, no other element was present?

No other elements were detected, probably due to their very low concentration.

-          The images from Figs. 2 and 3 should be provided at a better resolution.

New images have been provided

Reviewer 2 Report

This is a good manuscript but there are some flaws. In this work, the author investigated the effects of the anodization process on fibrobalst cellular responses and bacterial adhesion, and titanium surface was sued as the substrate. This study should provides some insights on effect of sruface composition and roughness on cell/bacterial adhesion. Some conerns need to be addressed before acceptance. 

1. Why was the manuscript titled with "gold-shaded surface modification"? I don't think gold was used in the surface modification.

2. Why MAC surface exhibits lowest protein and cell adhesion but highest biofilm formation, as the bioflim formation should be relevant to the protein/cell adhension.

3. Scale bar in Figure 1 is not readable. 

In Table 1, the titanium weight cocnetration of Ti "75.400" has a superscript "1". Please remove it.  

Author Response

1. Why was the manuscript titled with "gold-shaded surface modification"? I don't think gold was used in the surface modification.

We changed the title

2. Why MAC surface exhibits lowest protein and cell adhesion but highest biofilm formation, as the bioflim formation should be relevant to the protein/cell adhension.

We thank you for your punctual observation.
It is interesting to note that MAC exhibited a different behavior in terms of protein adsorption/cell
adhesion and biofilm formation with respect to the roughened surfaces. We think that it could be
since the roughness of the biomaterial, through differences in term of surface energy, could
differently impact on the interactions involving eucaryotic cells on one hand and bacteria on the
other. In this context, it is also important to note that the bacterial biofilm is mainly composed by
polysaccharides rather than proteins (key actors in mediating the interaction between cells and
surfaces). Finally, the biological assays we are referring to were performed with different timing
(minutes for adhesion and protein absorption versus 24 h for biofilm formation).

3. Scale bar in Figure 1 is not readable. 

The scale bar was enlarged

In Table 1, the titanium weight cocnetration of Ti "75.400" has a superscript "1". Please remove it.  

We have performed the requested correction

Round 2

Reviewer 1 Report

Thank you to the authors for addressing all my requests and observations.

Reviewer 2 Report

All my concerns are addressed and I believe this manuscript is now qualified for publication.